# The Roots of Rye (*Secale cereale* L.) Are Capable of Synthesizing Benzoxazinoids

**DOI:** 10.3390/ijms22094656

**Published:** 2021-04-28

**Authors:** Monika Rakoczy-Trojanowska, Bartosz M. Szabała, Elżbieta Różańska, Mariusz Kowalczyk, Wojciech Burza, Beata Bakera, Magdalena Święcicka

**Affiliations:** 1Department of Plant Genetics, Breeding and Biotechnology, Institute of Biology, Warsaw University of Life Sciences (SGGW), Nowoursynowska 166 St., 02-787 Warsaw, Poland; bartosz_szabala@sggw.edu.pl (B.M.S.); wojciech_burza@sggw.edu.pl (W.B.); beata_bakera@sggw.edu.pl (B.B.); magdalena_swiecicka@sggw.edu.pl (M.Ś.); 2Department of Botany, Institute of Biology, Warsaw University of Life Sciences (SGGW), Nowoursynowska 166 St., 02-787 Warsaw, Poland; elzbieta_rozanska@sggw.edu.pl; 3Department of Biochemistry and Crop Quality, Institute of Soil Science and Plant Cultivation-State Research Institute, Czartoryskich 8 St., 24-100 Puławy, Poland; mkowalczyk@iung.pulawy.pl

**Keywords:** *ScBx1* gene, indole-3-glycerol phosphate lyase, active compounds, seeds deprived of coleoptile, in vitro regeneration

## Abstract

According to current opinion, the first step of benzoxazinoids (BXs) synthesis, that is, the conversion of indole-3-glycerol phosphate to indole, occurs exclusively in the photosynthesising parts of plants. However, the results of our previous work and some other studies suggest that this process may also occur in the roots. In this study, we provide evidence that the first step of BXs synthesis does indeed occur in the roots of rye seedlings. We detected *ScBx1* transcripts, BX1 enzyme, and six BXs (2-hydroxy-1,4-benzoxazin-3-one, 2,4-dihydroxy-1,4-benzoxazin-3-one, (2***R***)-2-***O***-β-d-glucopyranosyl-4-hydroxy-(2***H***)-1,4-benzoxazin-3(4***H***)-one glucoside, 2,4-dihydroxy- 7-methoxy-1,4-benzoxazin-3-one, 2,4-dihydroxy-7-methoxy-1,4-benzoxazin-3-one glucoside, and 6-methoxy-2-benzoxazolinone) in the roots developed from seeds deprived of the coleoptile at 2 days after sowing (i.e., roots without contact with aerial parts). In roots regenerated in vitro, both *ScBx1* transcripts and BX1 enzyme were detected at a low but still measurable levels. Thus, BXs are able to be synthesised in both the roots and above-ground parts of rye plants.

## 1. Introduction

Benzoxazinoids (BXs), secondary metabolites synthesised by several species of the Poaceae family, play an important role in biotic and abiotic stress resistance, and in allelopathy. Common rye (*Secale cereale* L.) is among the species producing BXs at a particularly high level [1,2,3]. BX produced and secreted by rye roots makes it a suitable crop for sustainable weed control and organic farming [3].

There are several steps in BX biosynthesis, the first of which is the conversion of indole-3-glycerol phosphate to indole. Indole is transformed to indolin-1-one and then, after three subsequent mono-oxidations, 2,4-dihydroxy-1,4-benzoxazin-3-one (DIBOA) is synthesised. The glycosylation of DIBOA and 2,4-dihydroxy-7-methoxy-1,4-benzoxazin-3-one (DIMBOA) results in the production of (2R)-2-O-β-d-glucopyranosyl-4-hydroxy-(2H)-1,4-benzoxazin-3(4H)-one glucoside (GDIBOA) and 2,4-dihydroxy-7-methoxy-1,4-benzoxazin-3-one glucoside (GDIMBOA), respectively. The O-methylation of DIBOA and DIMBOA yields 2,4,7-trihydroxy-1,4-benzoxazin-3-one glucoside (GTRIBOA) and 2-hydroxy-4,7-dimethoxy-1,4-benzoxazin-3-one glucoside (GHDMBOA), respectively. Finally, hydroxylation reactions convert GDIBOA to DIBOA and GDIMBOA to DIMBOA [3,4,5,6,7].

The first step of BX biosynthesis is controlled by the *Bx1* gene (and its orthologs), which encodes the BX1 (benzoxazinless1) enzyme with the activity of indole-3-glycerol phosphate lyase [4,8,9,10].

In rye, besides *ScBx1* (Acc. No. KF636828.1), another gene encoding an indole-3-glycerol phosphate lyase, *ScIgl* (Acc. No. MN120476) takes over the role of *ScBx1* at later developmental stages—between the 42nd and 70th days after germination [11]. In maize, both enzymes can function efficiently in vitro [12,13].

According to current opinion, the conversion of indole-3-glycerol phosphate to indole takes place in the aerial parts of plants, specifically in the chloroplasts [4,5,6,7] and not in roots. However, the results of our previous experiments on rye BXs (2-hydroxy-1,4-benzoxazin-3-one—HBOA, DIBOA, GDIBOA, DIMBOA, GDIMBOA, and 6-methoxy-2-benzoxazolinone (MBOA)) and expression analyses of BX-related genes indicate that genes encoding indole-3-glycerol phosphate lyases, including *ScBx1*, are expressed both in the aerial parts and roots of 2- to 6-week-old seedlings. This was observed regardless of cultivation conditions, i.e., with or without Berseem clover [14], and at low or normal temperatures [15,16]. Similarly, Tanwir et al. [10] detected transcripts of six *ScBx* genes including *ScBx1* in the roots of germinating seeds and young seedlings of rye. Transcripts of *Bx1* genes have also been detected in the roots of other cereals, including wheat [17] and maize [18,19]. Nevertheless, none of the authors of the above cited works suggested that the first stage of BX biosynthesis may take place in the roots.

Although the results obtained by us and by other groups suggest that the first step of BX synthesis can occur in roots of at least three cereal species, no previous studies had presented experimental evidence confirming this. Moreover, none of the authors of the above cited works has even suggested such a possibility. Therefore, we decided to test the hypothesis that BXs can be synthesised in roots by conducting detailed analyses of roots developing without or with very short-lived contact with aerial plant parts. The results of these experiments and analyses are summarised in this paper.

## 2. Results

### 2.1. ScBx1 Gene Expression

Transcripts of *ScBx1* were detected in all analysed organs and organoids. In case of RDC, the transcript level of *ScBx1* was two times lower than that in R2W and 49.8 times lower than that in L2W. In RIV, the transcript level of *ScBx1* was seven times lower than that in RDC, 14 times lower than that in R2W, and nearly 350 times lower than that in L2W (Figure 1, Appendix A).

### 2.2. BX Content

Six BXs were detected in RDC. Two of the six analysed BXs (HBOA and especially DIBOA) were present at higher levels in RDC than in R2W: 1.56 and 2.18 times, respectively. Among all the organs compared, RDC had the highest DIMBOA content. The two other BXs—GDIBOA and GDIMBOA—were detected at lower levels in RDC than in R2W. The GDIMBOA level was 3.36 times higher in RDC than in L2W, and the GDIBOA level was 35.8 times lower in RDC than in L2W. The MBOA content was similar in RDC and R2W but was nearly 10 times higher in those organs than in L2W (Figure 2, Appendix A). The total BXs content was lower in RIV than in the other tested organs. GDIMBOA was either not produced in RIV, or its level was beneath the limit of detection. Among the BXs detected in RIV, GDIBOA, and DIMBOA were the most abundant, with the DIMBOA content comparable to that in L2W.

### 2.3. Bioinformatic Characterisation of BX1 Protein

The coding sequence of BX1 was 960 bp long, and encoded a protein of 319 amino acids with a calculated molecular weight of 33.7  kDa. The TargetP 2.0 program [20] predicted a chloroplast transit peptide of 41 amino acids at the N-terminus, indicating that the mature protein was approximately 29.3 kDa (Figure 3). The GRAVY index score of BX1 measured by the Kyte–Doolittle formula was above 0, suggesting that the protein may be hydrophobic and membrane-localised.

Database searches revealed that BX1 shared the highest identity (about 97%) with indole-3-glycerol phosphate lyases from *Triticum aeastivum* (Acc. No. KAF7049523), *Triticum turgidum* (Acc. No. VAI07221), and *Aegilops tauschii* (Acc. No. XP_020172362) (Figure 3), and lower identity with proteins from *Zea mays* (89%)*, Sorghum bicolor* (75%), *Digitaria exilis* (73%), *Oryza sativa* (72%), and *Secale cereale* (64%) (Acc. No. NP_001105219, XP_002463738, CAB3497862, XP_015633113, and QIB84921, respectively). The most divergent sequences were found within and after the putative chloroplast transit peptide.

### 2.4. Immunodetection of BX1 in the Plastid Fraction of Roots

First, the specificity of the affinity-purified anti-BX1 antibodies was determined in Western blotting analyses using total proteins extracted from roots of 2-week-old plants. The lack of any protein band indicated that BX1 was not present or its amount was below the limit of detection (Figure 4a).

Next, the plastids fractions isolated from RDC, R2W, and L2W were subjected to immunoblot analyses, which revealed a major band in the membrane fraction and a small band at approximately 29 kDa in the soluble fraction of plastids (Figure 4a). No signal was detected in the same protein extracts using pre-immune serum (Figure 4a). The molecular weight of the protein was highly consistent with the molecular weight calculated from the deduced amino acid sequence.

Relatively larger amounts of BX1 were detected in root plastids, from both RDC (especially) and R2W, than in leaf chloroplasts (Figure 4b). No signal was detected in the cytosolic fractions of any analysed organs.

The relative purity of isolated plastids and cytosol fractions was verified using compartment marker antibodies directed against FNR and UGpase (Figure 4b), respectively, and by light microscopy (Appendix A).

Sera raised against UDP-glucose pyrophosphorylase (UGPase) (cytosol marker) and ferredoxin NADP reductase FNR (plastid marker) were used in Western blots to confirm quality of isolated fractions.

### 2.5. Immunolocalisation of BX1 Protein in Plastids of Roots and Leaves

To complement the cell fractionation results described above, we conducted immunogold labelling and observed the localisation of labelled components by transmission electron microscopy. The detection of BX1 on an ultrastructural level confirmed its specific localisation in plastids of RDC (Figure 5) as well as in both controls-plastids of R2W and chloroplasts of L2W (Appendix A).

## 3. Discussion

Many studies on different aspects BX biosynthesis over the past 50 years have reported that this class of metabolites is present in roots, root exudates, or root extracts of such species as maize [6,21,22,23], common wheat [24,25,26,27], emmer wheat [28], and rye [2,14,15,16,24,29]. Additionally, transcripts of *Bx1*, the first gene controlling the BX biosynthesis pathway, have been detected in roots of rye [10,14,15,16], wheat [17], and maize [18,19].

Recently, we proved that one more gene, *ScIgl*, controls the first step of BX biosynthesis at later developmental stages [11] and that this gene is expressed in roots of rye [14]. The results of qRT-PCR analyses of *ScIgl*, which showed the highest transcript levels in RDC and R2W, are presented in Appendix A.

According to current opinion, the first step in BX biosynthesis, i.e., the conversion of indole-3-glycerol phosphate to indole by BX1, occurs in the aerial parts of plants, specifically in chloroplasts. Some authors, e.g., Tanwir et al. [10], have postulated that BXs are mainly biosynthesised in rye shoots, not in roots, and then transported to the roots where they accumulate as methoxylated glycosylated forms. This point of view is difficult to reconcile with the fact that *ScBx1* (as well as its “co-operator”, *ScIgl*) is expressed in roots at low but still detectable levels, regardless of cultivation conditions and the plant age and genotype [14]. At the same time, the detection of *ScBx1* transcripts and BXs in roots was convincing, but not unequivocal, evidence that the first stage of BX biosynthesis may occur in both aerial parts and roots of rye. Therefore, in this study, we aimed to obtain definitive evidence to show that BX1 is present and active in the roots of rye. This evidence was obtained by performing biochemical, gene expression, and immunological assays of “independent roots”, i.e., those developed from seeds deprived of the coleoptile immediately after emergence, so that they had no contact with the photosynthesising aerial plant parts. To provide irrefutable evidence showing that the biosynthesis of BXs can initiate and continue independently in the roots, we conducted gene expression and biochemical analyses of roots developed in vitro. These organoids have never come into contact with aerial parts, so we could be sure that any BXs detected in the roots had not been transported from the aerial plants.

To date, no previous studies have focused on the course of BX biosynthesis in rye roots, because the main focus has been on metabolites secreted by roots and their influence on nematodes [29,30,31] and weeds [2,32,33]. To the best of our knowledge, no studies have focused on BX biosynthesis in the roots of other species that produce BXs, either.

We detected *ScBx1* transcripts in all four analysed organs—RDC, RIV, R2W, and L2W. As expected, the transcript levels were much lower in roots, particularly RIV, than in leaves of 2-week-old seedlings. Relatively lower transcript levels of *ScBx1* in roots than in aerial parts at early developmental stages (48 h, 72 h, 2 days, 6 days, and 8 days after seed germination) were also reported by Tanwir et al. [10]. The same phenomenon, regardless of cultivation conditions (with or without Berseem clover), was observed by Rakoczy-Trojanowska et al. [14]. However, the differences in *ScBx1* transcript levels between RDC with R2W were minor.

We detected BXs in almost all analysed tissues (except for GDIMBOA in RIV), with the lowest levels detected in RIV. As expected from the organ-specific nature of BX biosynthesis, the contents of methoxylated BX metabolites (MBOA, DIMBOA, and GDIMBOA) were higher in roots (both RDC and R2W) than in leaves (L2W), whereas the contents of non-methoxylated BX metabolites were higher in leaves than in roots. The same pattern of distribution was detected by Tanwir et al. [10] and Rakoczy-Trojanowska et al. [14]. In the latter study, the distribution pattern was the same in rye plants cultivated with and without Berseem clover.

Interestingly, in the present study, the concentrations of DIBOA and DIMBOA, two key BXs in rye, were more than two times higher, and that of HBOA was more than 1.5 times higher, in RDC than in R2W. These high concentrations may be because they could not be transported to the aerial parts. During typical plant development, different compounds, including BXs, are transported via the xylem from roots to the upper parts [34,35,36,37]. Recently, Hazrati et al. [38] showed that two BXs, GDIBOA and GDIMBOA, were secreted by rye roots into the soil and then absorbed by the roots of the co-cultivated hairy vetch plants and translocated to their shoots. However, when the contact between the roots and the upper parts is broken, e.g., by removal of the coleoptiles, metabolites cannot be transported and so they accumulate in excessive amounts at the site of their synthesis, in this case, in the roots. In addition, the pool of BXs in RDC may have increased in response to wounding during coleoptile removal, a factor that has been shown to stimulate BX production [8]. The contents of five out of six BXs were lower in RIV than in the other tested organs: as much as seven times lower than in RDC. The other BX, GDIMBOA, was undetectable rather than absent (because GDIBOA was detected). Plant tissues, especially callus and roots, are capable of producing secondary metabolites [39,40]. Therefore, we expected that the concentrations of BXs in RIV would be relatively high. The fact that we detected lower contents of most BXs in RIV than in other organs may be due to the in vitro culture conditions; that is, those that induce rhizogenesis, rather than those that induce BX production.

Tanwir et al. [10] postulated that non-methoxylated BXs (HBOA and DIBOA) were transported to the roots even before removing the coleoptiles. However, this is contradicted by our findings that the contents of both non-methoxylated BXs were higher in RDC than in R2W at a similar stage of development (1.6 times and 2.2 times higher for HBOA and DIBOA, respectively). If the BXs produced in coleoptiles were transported to roots, then the amount would be very small: firstly, because of the early developmental stage (BX synthesis is low in coleoptiles during the first 2 days); secondly, because of the very short contact time between shoots and roots (thus, a short time available for BX transport); and thirdly, because of the long time between removing the coleoptiles and the sampling time. This possibility is obviously completely excluded in the case of RIV (roots regenerated in vitro).

Both immunoassays used in our study detected BX1 in plastids of the roots, including RDC, and in the leaves. Although other studies have detected *Bx1* transcripts in roots, there are no previous reports on the presence and the role of BX1 protein in root plastids. In this study, we report a new observation that has not been previously reported for other lyases. BX1 catalysing indole production has a signal peptide for plastid import, encoded by the first exon of its encoding gene [41,42], so its plastid subcellular localisation was expected. Interestingly, we detected the highest levels of BX1 protein in RDC, medium levels in R2W, and the lowest levels in L2W, opposite to the trends in *ScBx1* transcript levels. A poor correlation between mRNA levels and protein levels in eukaryotes, where the two processes are spatially separated, is a well-known phenomenon that has been reported by many authors (for reviews see: [43,44,45]). It can be assumed that in RDC, which develop under conditions of permanent stress, a limited amount of mRNA is used more effectively than the larger amounts of mRNA in the other two organs. Upon environmental challenges, e.g., in roots developing atypically without contact with photosynthesising upper plant parts, translation would be regulated to reduce energy consumption, as noted by Echevarría-Zomeño et al. [46]. All this might imply a highly efficient enzymatic reaction of the conversion of indole to indolin-1-one in RDC. Moreover, the course and specificity of post-transcriptional regulatory processes (e.g., formation of alternative splicing variants resulting from intron retention and alternative 5′ and 3′ splice sites [47]) may explain the weak relationship between the observed transcript and protein levels. Our results highlight that transcriptomic data should be interpreted with caution and should not be the basis for far-reaching conclusions.

The presence of BX1 in RDC, R2W, and L2W is further evidence that BX biosynthesis can occur independently in roots and leaves. Moreover, the higher levels of BX1 protein in R2W and especially in RDC indicate high efficiency of the enzymatic reaction in roots (particularly in roots developing independently without contact with photosynthesising upper plant parts).

On the basis of the results of all our analyses, that is, the detectable transcript levels of *ScBx1*, high contents of three BXs (HBOA, DIBOA, and DIMBOA), and large amounts of BX1 protein in RDC, we conclude that BXs are synthesised de novo in roots, independently of aerial plant parts. Our results do not exclude the possibility that some BXs are transported from the aerial parts to the roots. Therefore, we would like to update the currently accepted opinion that the first stage occurs only in chloroplasts; on the contrary, it can also take place in roots, at least in rye. Our results not only broaden knowledge of BX biosynthesis but also may have potential practical applications. The fact that BXs (including those that function in defence, such as DIBOA and DIMBOA) are synthesised in roots without contact with the aerial parts, especially roots regenerated in vitro, opens opportunities to produce these compounds in roots and, very likely, in hairy root in vitro cultures. This will require the development of appropriate methodologies including appropriate stimulators, media, internal and external in vitro culture conditions, and biotic and abiotic elicitors, as suggested by Espinosa-Leal et al. [39] and Chandran et al. [48], among others.

## 4. Materials and Methods

### 4.1. Plant Material

The following organs of rye (*S. cereale* L.) inbred line L318 (S37), bred in the Department of Plant Genetics, Breeding and Biotechnology, were used in analyses:-roots developed from seeds deprived of the coleoptile at 2 days after sowing (RDC),-roots developed in vitro (RIV),-roots (R2W) and leaves (L2W) from 2-week-old seedlings.

R2W and L2W were treated as control reference organs (designated as KI and KII, respectively).

To obtain RDC, 180 seeds were sown in Petri dishes on wet filter paper and left to germinate in the dark for 1–2 days at 24 °C. Out of 180 seeds, 118 germinated. The emerging coleoptiles were cut from the seedlings immediately after emergence and seeds with growing roots were placed back into darkness for another 10 days (Appendix A). After this period, RDC were divided into four portions for gene expression, biochemical, immunological, and microscopy analyses.

To produce RIV, an in vitro culture was established from mature embryos. Before embryo isolation, seeds were sterilised in a 3% solution of sodium hypochlorite followed by immersion in 70% ethyl alcohol for 3 min. The seeds were then washed three times with sterile water (for 15, 10, and 5 min). Sterile embryos were placed individually in Petri dishes (Φ 10 cm) on a thin layer of modified MS medium (half strength of macro-elements, 30 g/dm^3^ sucrose). Roots formed after 2–3 weeks were separated from primary explants, cut into fragments of 0.5–1 cm, and transferred to Erlenmeyer flasks containing 40 mL of the same medium (0.5 PCV) with Picloram at a concentration of 3 mg/dm^3^. The cultures were kept under a 16-h light/8-h dark photoperiod (photosynthetic photon flux density = 50 μmol m^−2^ s^−1^) at 25 °C. After 10 days, secondary roots had formed. The subcultures were done every 7–10 days. Roots (hereinafter referred to as RIV, or more generally organoids) used for gene expression and biochemical analyses were collected between the 14th and 21st day after culture initiation.

The R2W and L2W samples were collected from 2-week-old seedlings. First, seeds were sown in Petri dishes and after 2 days, germinating seeds were transferred to pots containing a 1:1 mixture of perlite and peat substrate (peat mixed with chalk—6 kg/m^3^, MAKRO MIS4 fertiliser—2 kg/m^3^, and MIKRO MIS fertiliser—100 g/m^3^) and cultivated under a 12-h light/12-h dark photoperiod (15 °C days, 12 °C nights) and 60%–80% relative humidity. This experiment was performed with four biological replicates of 21 plants per replicate.

Immediately after sampling, all collected organs and organoids were frozen in liquid nitrogen, and tissues intended for use in biochemical analyses were lyophilised.

### 4.2. Bioinformatics Analysis of BX1

The TargetP 2.0 program [20] was used to predict the transit peptide. The GRAVY index (average hydrophobicity) was determined according to Kyte and Doolittle [49]. Multiple sequence alignment of different lyases was performed using the Clustal Omega sequence alignment program (www.clustal.org; accessed on 18 August 2020).

### 4.3. RNA Isolation, cDNA Synthesis, and Quantitative real Time-PCR Analysis

A GeneMATRIX Universal RNA Purification Kit (version 1.2) (Eurx, Gdańsk, Poland) was used to extract total RNA from RDC, RIV, L2W, and R2W. The isolated RNA was dissolved in 40 μL RNase-free water. The RNA integrity and concentration were measured with a NanoDrop 2000 spectrophotometer. Then, the RNA was treated with Turbo DNase (Thermo Fisher Scientific, Waltham, MA, USA) according to the manufacturer’s instructions. The cDNA was transcribed using a High-Capacity cDNA Reverse Transcription kit (Thermo Fisher Scientific) and diluted with RNase-free water.

The transcript levels of *ScBx1* were determined by qRT-PCR. The experiments were carried out in 96-well reaction plates with a Light Cycler 96 Instrument (Roche, Basel, Switzerland). The reaction conditions were as follows: 35 cycles at 95 °C for 10 s, 57 °C for 10 s, and 72 °C for 15 s. *HvAct* (Acc. No. AY145451) served as the internal control and was chosen based on the geNorm algorithm (https://genorm.cmgg.be; accessed on 4 November 2020). This gene was chosen as the most stable based on preliminary experiments on *ScGADPH* (Acc. No. JQ659189.1) and *Ta54227* (Acc. No. Pr0102692400; [50]). The reaction mixture (total volume 20 µL) consisted of 4 μL cDNA (5 ng/μL), 1 μL each gene-specific primer (5 mM), 4 μL RNase-free water, and 10 μL FastStart Essential DNA Green Master (Roche). The transcript level of the target gene was normalised to that of *HvAct* by the 2^−ΔΔCt^ method [51]. Table 1 shows the sequences of the primers used to amplify *ScBx1* and *HvAct*.

### 4.4. Biochemical Analysis

Previously published protocols [11,52] with some modifications were used to quantify the following BXs: HBOA, GDIBOA, DIBOA, GDIMBOA, DIMBOA, and MBOA. Briefly, plant material (100 mg) was extracted with 70% methanol containing an internal standard (2 µg/mL indoxyl β-D-glucoside; IbG) under the conditions described previously. The obtained extracts were evaporated to dryness under reduced pressure, reconstituted in 1 mL methanol containing 0.1% (*v*/*v*) acetic acid, and stored at −20 °C. Before analysis, samples were centrifuged for 20 min at 23,000× *g* at 4 °C and filtered using Whatmann Mini-Uniprep filtering vials (GE Healthcare, Buckinghamshire, UK) with a regenerated cellulose membrane (0.2-µm pore size).

The BXs were separated on a Waters BEH C18 column (100 × 2.1 mm, 1.7 µm, Waters, Milford, MA, USA) with a linear, 8.5-min gradient from 3% to 15% acetonitrile containing 0.1% (*v*/*v*) formic acid (solvent B) in 0.1% formic acid (solvent A). All separations were carried out on a Waters Acquity UPLC system with the column temperature set to 50 °C and a flow rate of 0.6 mL/min.

The column effluent was redirected into the waste for the first 1.5 min of separation, to prevent contamination of the ion source with polar constituents of the sample matrix. The effluent was subsequently directed into a triple quadrupole mass spectrometer (Waters TQD, Waters, Milford, MA, USA) and analysed as described previously [11,14,52]. Data acquisition and processing were conducted using Waters MassLynx 4.1 SCN 919 software (Waters, Milford, MA, USA).

The calibration used seven concentration points, between 0.3 and 35 ng/µL, prepared from standard solutions of DIBOA, DIMBOA, HBOA, GDIMBOA, MBOA, and IbG at 1 mg/mL each. The GDIMBOA curve was used for calibration of GDIBOA. Each concentration point response was calculated as the ratio of the analyte to the IgG peak area. For all investigated compounds, the relationship of concentration vs. response was linear up to 30 ng/µL. The accuracy of the quantification was monitored by injecting quality control samples, consisting of a mixture of all standards at 1, 15, and 30 ng/μL, every ten injections of plant extract samples. Method detection limits were determined from ten separate analyses of 0.3 ng/µL solutions of the calibration standards.

### 4.5. Immunodetection of BX1 Protein in Root and Leaf Plastids

#### 4.5.1. Preparation of Antiserum

Polyclonal antibodies against the peptide VTGPRENVNLRVESL derived from the protein sequence of the rye chloroplast BX1 were raised in a rabbit at Agrisera company (Vännäs, Sweden; https://www.agrisera.com). The antiserum was affinity-purified using the peptide antigen coupled to UltraLink^TM^ Iodoacetyl gel (Thermo Fisher Scientific).

Preimmune serum was affinity-purified using a Protein A IgG Purification Kit (Pierce, Rockford, IL, USA) following the manufacturer’s instructions, and was used as a negative control in Western blot and histochemical analyses.

#### 4.5.2. Plastid Isolation

Plastids were isolated from roots (RDC and R2W) and leaves (L2W). Plant material was ground in a mortar and pestle followed by homogenisation in a Waring blender for 2 × 10 s in ice-cold buffer (0.33 M sorbitol, 50 mM Tris–HCl, pH 8.0, 2 mM EDTA, 0.1% (*w*/*v*) bovine serum albumin (BSA), and 5 mM ascorbic acid). The homogenate was filtered through 100-µm and 40-µm nylon mesh and centrifuged at 200× *g* for 2 min and at 2000× *g* for 7 min. The resulting pellet of enriched plastids was washed twice in homogenisation medium and then subjected to Western blot analysis.

The quality of the isolated plastid fraction was verified by light microscopy and Western blotting using compartment marker antibodies directed against ferredoxin NADP reductase (FNR) (Agrisera).

#### 4.5.3. Cytosol Isolation

The cytosolic fraction was isolated from roots (RDC and R2W) and leaves (L2W). Plant material was ground in a mortar and pestle and re-suspended in ice-cold homogenisation medium (0.4 M sucrose, 50 mM Tris–HCl, pH 8.0, 5 mM EDTA, 0.1% PVP, 0.1% (*w*/*v*) BSA, 5 mM ascorbic acid, and protease inhibitor cocktail). The homogenate was filtered through 100-µm nylon mesh and centrifuged at 12,000× *g* for 20 min. The resulting supernatant was used as the cytosolic fraction in Western blot analyses. The quality of the isolated cytosolic fraction was verified by Western blotting using compartment marker antibodies directed against UDP-glucose pyrophosphorylase (Agrisera).

#### 4.5.4. Preparation of Protein Extracts

Soluble proteins were extracted from isolated organelles in 50 mM Tris-HCl pH 8.0, 5 mM EDTA, 5% β-mercaptoethanol, and protease inhibitor cocktail. After removal of the soluble fraction, membrane proteins were extracted in the same buffer except that it contained 2% SDS. The protein content was determined using a Modified Lowry Protein Assay Kit (Thermo Fisher Scientific).

#### 4.5.5. Western Blot Analysis

Proteins (10 µg) were separated by 10–12.5% SDS-PAGE and electroblotted onto a 45-µm nitrocellulose membrane (GE Healthcare). Membranes were stained with Ponceau to ensure equal distribution of proteins in each lane. After blocking in a mixture of tris-buffered saline and Polysorbate 20 containing 5% (*w*/*v*) non-fat dry milk powder, the membrane was incubated with the primary antibody diluted 1:2000 for 1 h. Following incubation and washing steps, the membrane was incubated with secondary goat anti-rabbit antibody HRP conjugated (Agrisera) for 1 h diluted 1:40000. The protein-antibody complex was detected with the ECL Superbright system (Agrisera) and photographed using the ChemiDoc Imaging System (Bio-Rad, Hercules, CA, USA). Each experiment was performed in triplicate.

### 4.6. Detection of BX1 Protein by Transmission Electron Microscopy

Segments of RDC, R2W, and L2W were fixed in 2% (*w*/*v*) paraformaldehyde (Sigma-Aldrich, St. Louis, MO, USA) and 2% (*w*/*w*) glutaraldehyde (Sigma-Aldrich) in 10 mM phosphate-buffered saline (PBS) buffer pH 7.4 for 2 h. Samples were dehydrated with increasing concentrations of ethanol, and then infiltrated and embedded in epoxy resin (Sigma-Aldrich; equivalent to Epon 812) according to the manufacturer’s instructions. The specimens were polymerised at 56 °C for 3 days [53]. Immunogold labelling for transmission electron microscopy analyses was conducted on sections obtained from the same samples (collected in three independent experiments and at least five randomly selected specimens) using the same 3-µm thick sections examined by light microscopy. Ultrathin sections (90 nm) were made for transmission electron microscopy with a Leica UCT ultramicrotome (Leica Microsystems, Wetzlar, Germany) and placed on nickel grids. Immunogold labelling was performed on grid-mounted thin sections floating in drops, according to Brorson et al. [53], with modifications. First, ultrathin epoxy sections were treated with 15% H_2_O_2_ for 10 min, and then rinsed three times for 5 min in MQ water and pre-incubated in 10 mM PBS for 10 min. Sections were incubated in 4% (*w*/*v*) BSA diluted in 10 mM PBS (pH 7.2) for 2 h before immunolabeling to block non-specific labelling. The primary purified polyclonal antibody (rabbit Anti-IGPPL, Agrisera) described above was diluted 1:50 in 4% (*w*/*v*) BSA and then incubated with the grids 1.5 h at room temperature. Samples were washed in 10 mM PBS with 0.05% (*v*/*v*) Tween 20 four times for 10 min and once in 10 mM PBS for 10 min. The secondary antibody (goat anti rabbit conjugated to 18 nm colloidal gold; Jackson ImmunoResearch, Ely, UK) was diluted 1:100 in 4% (*w*/*v*) BSA and incubated with the grids for 1 h at room temperature. The grids were then washed three times in 10 mM PBS with 0.05% (*v*/*v*) Tween 20 and rinsed five times in MQ water. The grids were stained with 1.2% (*w*/*v*) uranyl acetate solution in 70% methanol (Sigma-Aldrich) for 1 min and washed five times with MQ water, each for 2 min. In negative controls, the primary antibody was omitted. Samples were examined under an FEI 268D “Morgagni” (FEI Company, Hillsboro, OR, USA) transmission electron microscope operating at 80 kV. Images were acquired with an SIS “Morada” digital camera (Olympus-SIS, Münster, Germany).

## Figures and Tables

**Figure 1 ijms-22-04656-f001:**
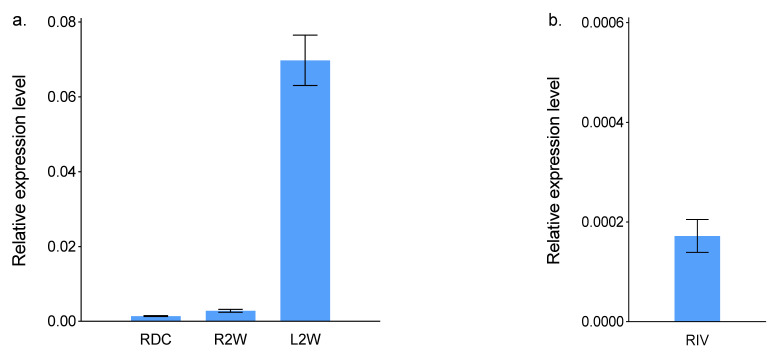
Relative transcript levels of *ScBx1* in (**a**) roots developed from seeds deprived of coleoptile 2 days after sowing (RDC), roots (R2W) and leaves (L2W) of 2-week-old seedlings; (**b**) roots developed in vitro (RIV) of rye inbred line L318. Data are mean ± SD of three biological replicates.

**Figure 2 ijms-22-04656-f002:**
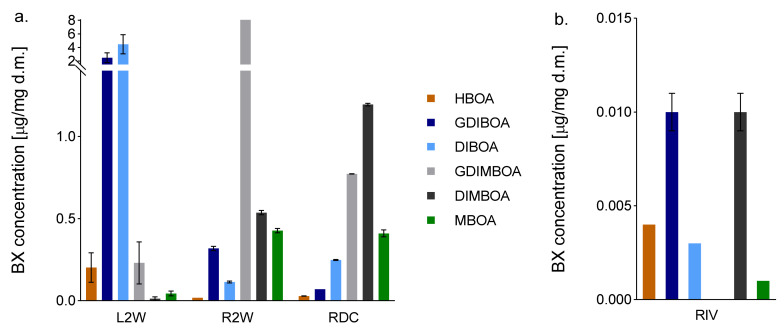
Concentrations of benzoxazinoids (BXs) in roots developed from (**a**) seeds deprived of coleoptile 2 days after sowing (RDC), roots (R2W) and leaves (L2W) of 2-week-old seedlings; and (**b**) roots developed in vitro (RIV) of rye inbred line L318. Data mean ±SD of three biological replicates.

**Figure 3 ijms-22-04656-f003:**
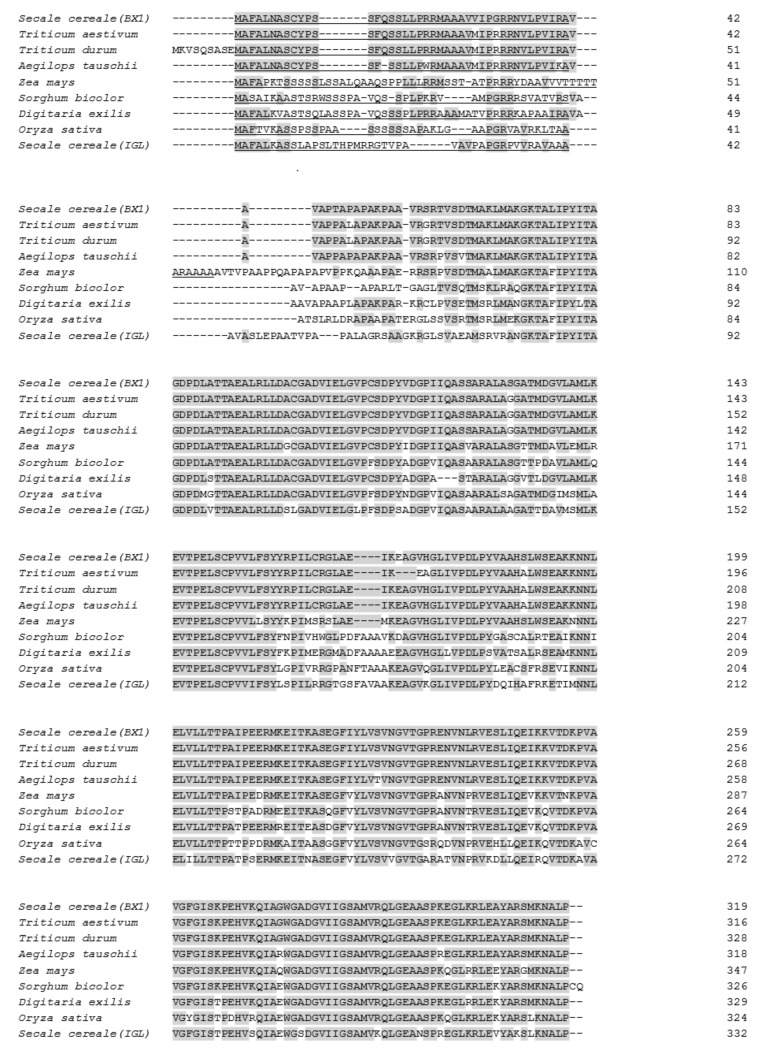
Amino acid sequence alignment of BX1 and other lyases from *Triticum aestivum* (Acc. No. KAF7049523), *Triticum turgidum* (Acc. No. VAI07221), *Aegilops tauschii* (Acc. No. XP_020172362), *Zea Mays* (Acc. No. NP_001105219) *Sorghum bicolor* (Acc. No. XP_002463738), *Digitaria exilis* (Acc. No. CAB3497862), *Oryza sativa* (Acc. No. XP_015633113), and *Secale cereale* (Acc. No. QIB84921). Residues identical between BX1 and all other proteins are shaded. Putative chloroplast transit peptides of proteins as predicted by the TargetP 2.0 are underlined.

**Figure 4 ijms-22-04656-f004:**
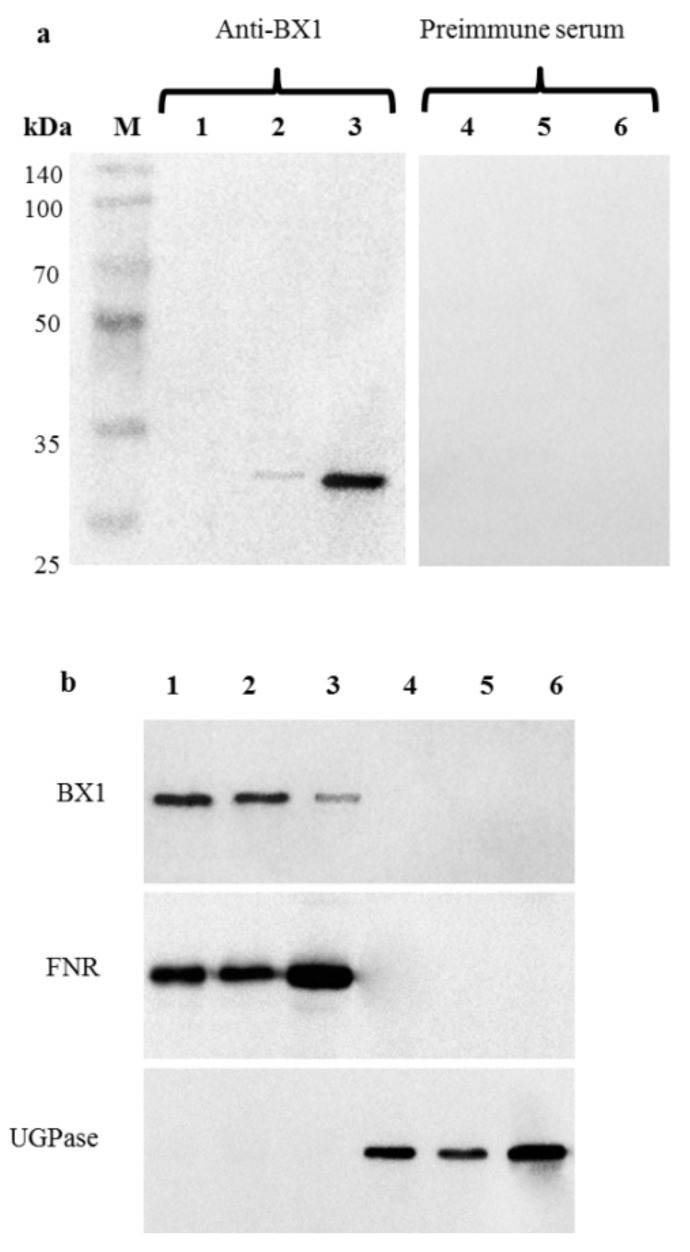
Immunoblot analysis of BX1 in different extracts of roots developed from seeds deprived of coleoptile 2 days after sowing (RDC), roots (R2W) and leaves (L2W) of 2-week-old seedlings of rye inbred line L318. (**a**) Immunospecificity of affinity-purified anti-BX1 polyclonal antibodies in root protein extracts (RDC): (1,4) Total proteins from roots; (2,5) Soluble proteins from root plastids; (3,6) Membrane proteins from root plastids; (M) Molecular weight marker (Eurx, Poland). (**b**) Detection of BX1 in plastid and cytosol fractions of different organs: (1) Total fraction of plastids isolated from roots developed from seeds deprived of coleoptile 2 days after sowing (RDC); (2) Total fraction of plastids isolated from roots of 2-week-old seedlings (R2W); (3) Total fraction of chloroplasts isolated from leaves of 2-week-old seedlings (L2W); (4) Cytosolic fraction isolated from RDC; (5) Cytosolic fraction isolated from R2W; (6) Cytosolic fraction isolated from L2W.

**Figure 5 ijms-22-04656-f005:**
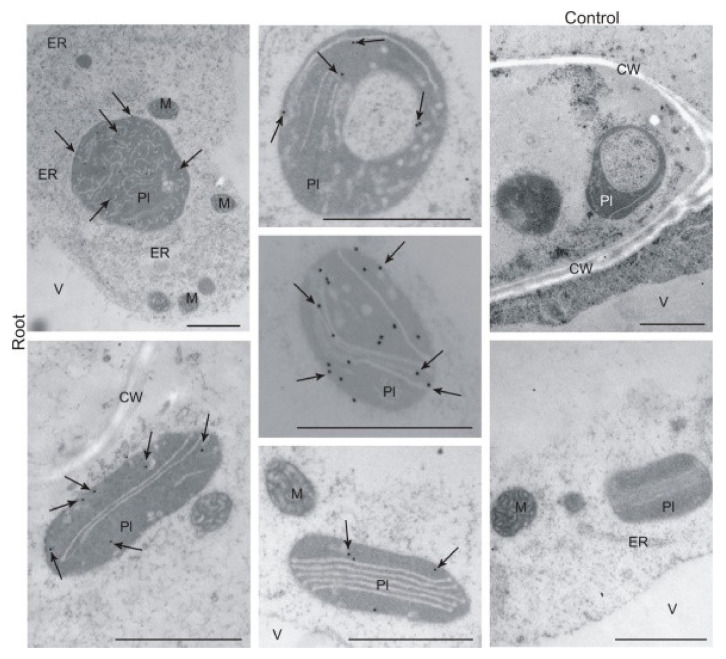
Localisation of BX1 by immunogold labelling and transmission microscopy in roots developed from seeds deprived of coleoptile 2 days after sowing (RDC) of rye inbred line L318. Abbreviations: CW—cell wall, ER—endoplasmic reticulum, M—mitochondrion, Pl—plastid, V—vacuole. Scale bars = 1 µm.

**Table 1 ijms-22-04656-t001:** Primers used in the qRT-PCR reactions.

Gene	Sequences (5′–3′)
*ScBx1*	F: TCAAAACCTGAACACGTGAAGC
	R: GCCTCTAGCCTTTTCAATCCTTC
*HvAct*	F: CCCCTTTGAACCCAAAAGCC
	R: GAAAGCACGGCCTGAATAGC

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
