# Peer review of "The Roots of Rye (Secale cereale L.) Are Capable of Synthesizing Benzoxazinoids"

_ijms, 2021, doi:10.3390/ijms22094656_

Round 1
Reviewer 1 Report
The article “The roots of rye (Secale cereale L.) are capable of synthesizing benzoxazinoids” deserves reader’s attention, and will especially be of interest to those working with rye and other cereals. Authors did a comprehensive work and the results bring a new understanding of benzoxazinoids synthesis, regarding the capability to be synthesized in rye root without participation of aerial parts of the plant. That statement is in some contradiction with a previous knowledge, but authors do very well reinforcing their point of view by a variety of results obtained by a range of methods: expression data for BX1 gene, biochemical analysis of presence Bxs, immunodetection of BX1 protein and immunolocalization of BX1 protein in plastids of roots and leaves. The results are very convincing and leaving no room for doubts. Perhaps, in future research it will be also possible to analyze the whole biochemical pathway of benzoxazinoids synthesis (including all BX genes, not only BX1) in the root tissue and to understand the basic of the differences in benzoxazinoids synthesis in roots and leaves.
There are only few small comments to the work.
Lines 44-45: The first step of BX biosynthesis is controlled by the Bx1 gene (and its orthologs), which encodes BX1 (benzoxazinless1) enzyme/protein? with the activity of indole-3-glycerol phosphate lyase [4,8-10].
Lines 234-236: If the BXs produced in coleoptiles were transported to roots, then the amount would be very small: firstly, because of the early developmental stage (BX synthesis is low in coleoptiles during the first 2 days);… There is no data in the article regarding BX synthesis in coleoptiles. Is it some previously published data without a reference in the text?
Line 324: sequence alignment program (www.clustal.,org) (no comma is needed)
Lines 362-363: The effluent was subsequently directed into a triple quadrupole mass spectrometer 362(Waters TQD, Manchester, UK) and analysed as described previously. “Previously” – you mean protocols from references [11,52]?
Line 374: 4.5. Immunodetection of BX1protein in root and leaf plastids (space between words is missing)

Author Response
Responses to Rev 1 comments
We would like to thank the reviewers for their critical comments, most of them were taken into account when improving the manuscript.
Lines 44-45: The first step of BX biosynthesis is controlled by the Bx1 gene (and its orthologs), which encodes BX1 (benzoxazinless1) enzyme/protein with the activity of indole-3-glycerol phosphate lyase [4,8-10].
After the parentheses, we have added the word “enzyme”
Lines 234-236: If the BXs produced in coleoptiles were transported to roots, then the amount would be very small: firstly, because of the early developmental stage (BX synthesis is low in coleoptiles during the first 2 days);… There is no data in the article regarding BX synthesis in coleoptiles. Is it some previously published data without a reference in the text?
In the paragraph 230-239 we discuss (with skepticism) results the work of Tanwir et al. (2017) who postulated that HBOA and DIBOA are only transported to the roots from the coleoptiles because we have proved that they are synthesized in roots de novo. Then, we present a hypothetical scenario – what if in 2-day-old coleoptiles, before their removal, BX were synthesized and transported to the roots. There is no data in the article regarding BX synthesis in coleoptiles because, we intentionally did not analyzed the BX content in the removed coleoptiles.
The work of Tanwir et al. (2017) is the only one with which we can compare our results.
Line 324: sequence alignment program (www.clustal.,org) (no comma is needed)
done
Lines 362-363: The effluent was subsequently directed into a triple quadrupole mass spectrometer 362(Waters TQD, Manchester, UK) and analysed as described previously. “Previously” – you mean protocols from references [11,52]?
“Previously” – we mean protocols from references 11,14 and 52; we have added these references after the word “previously”.?
Line 374: 4.5. Immunodetection of BX1 protein in root and leaf plastids (space between words is missing)
done

Reviewer 2 Report
I read with interest the manuscript "The roots of rye (Secale cereale L.) are capable of synthesizing 2 benzoxazinoids". In general, the manuscript is well written and throws novel light on the synthesis of benzoxazinoids in plants, at least in rye. The discussion is very nicely presented. However, I suggest the authors improving the introduction, not restricting it only on the synthesis of benzoxazinoids (even though the ms is focused on this topic), but on other aspects of them in order to strengthen the importance of this study.
I would also ask the authors for adding a section of statistical analysis of data in the main text, with statistics clearly evident in the graphs.
Minor remark: the legend of Figure 2 seems to lack part of its description. Please, check.
Author Response
Responses to Rev 2 comments
We would like to thank the reviewers for their critical comments, most of them were taken into account when improving the manuscript.
I read with interest the manuscript "The roots of rye (Secale cereale L.) are capable of synthesizing 2 benzoxazinoids". In general, the manuscript is well written and throws novel light on the synthesis of benzoxazinoids in plants, at least in rye. The discussion is very nicely presented. However, I suggest the authors improving the introduction, not restricting it only on the synthesis of benzoxazinoids (even though the ms is focused on this topic), but on other aspects of them in order to strengthen the importance of this study.
We prefer a short and to the point introduction (we use this form in all our articles) and we would not like to enlarge the content of this section too much. We think that the importance of the research is best emphasized by the discussion and we tried to write this section as fully as possible. However, sharing Rev's view that Introduction should be helpful in strengthening the importance obtained results – we have added some sentences and we changed the wording of a few sentences a bit.
I would also ask the authors for adding a section of statistical analysis of data in the main text, with statistics clearly evident in the graphs.
We did not perform any sophisticated statistical tests because the differences between mean values of gene expression and BX content for the compared tissues were evident. The only one statistics applied is standard deviation reflecting the variability between biological replicates. Therefore, we think that it makes no sense to add of a separate section for statistical analysis. However, if Rev. 2 still deems it necessary - we will add such a section.
Minor remark: the legend of Figure 2 seems to lack part of its description. Please, check.
Done, now the description is: ”Concentrations of benzoxazinoids (BXs) in roots developed from (a) seeds deprived of coleoptile 2 days after sowing (RDC), roots (R2W) and leaves (L2W) of 2-week-old seedlings; and (b) roots developed in vitro (RIV) of rye inbred line L318. Data mean ± sd of three biological replicates.”
